# Biomimetic Inorganic Nanovectors as Tumor-Targeting Theranostic Platform against Triple-Negative Breast Cancer

**DOI:** 10.3390/pharmaceutics15102507

**Published:** 2023-10-22

**Authors:** Huang Wen, Pekka Poutiainen, Enkhzaya Batnasan, Leena Latonen, Vesa-Pekka Lehto, Wujun Xu

**Affiliations:** 1Department of Technical Physics, University of Eastern Finland, Yliopistonranta 1F, 70211 Kuopio, Finland; hwen@uef.fi; 2Kuopio University Hospital, University of Eastern Finland, Puijonlaaksontie 2, 70210 Kuopio, Finland; pekka.poutiainen@pshyvinvointialue.fi; 3School of Medicine, University of Eastern Finland, Yliopistonranta 1F, 70211 Kuopio, Finland; enkhzaya.batnasan@uef.fi (E.B.); leena.latonen@uef.fi (L.L.)

**Keywords:** mesoporous silicon nanoparticles, bisphosphonates, cancer cell membrane coating, radiolabeling, doxorubicin, triple-negative breast cancer

## Abstract

Mesoporous silicon nanoparticles (PSi NPs) are promising platforms of nanomedicine because of their good compatibility, high payload capacities of anticancer drugs, and easy chemical modification. Here, PSi surfaces were functionalized with bisphosphonates (BP) for radiolabeling, loaded with doxorubicin (DOX) for chemotherapy, and the NPs were coated with cancer cell membrane (CCm) for homotypic cancer targeting. To enhance the CCm coating, the NP surfaces were covered with polyethylene glycol prior to the CCm coating. The effects of the BP amount and pH conditions on the radiolabeling efficacy were studied. The maximum BP was (2.27 wt%) on the PSi surfaces, and higher radiochemical yields were obtained for ^99m^Tc (97% ± 2%) and ^68^Ga (94.6% ± 0.2%) under optimized pH conditions (pH = 5). The biomimetic NPs exhibited a good radiochemical and colloidal stability in phosphate-buffered saline and cell medium. In vitro studies demonstrated that the biomimetic NPs exhibited an enhanced cellular uptake and increased delivery of DOX to cancer cells, resulting in better chemotherapy than free DOX or pure NPs. Altogether, these findings indicate the potential of the developed platform for cancer treatment and diagnosis.

## 1. Introduction

Cancer is one of the leading causes of death worldwide due to the poor specificity, inadequate drug distribution, and severe systemic toxicity of conventional therapy approaches such as chemotherapy, radiotherapy, and surgery [1,2]. In particular, triple-negative breast cancer (TNBC), with a poor prognosis and high probability of recurrence and metastasis, is one of the most aggressive cancers [3]. Moreover, the lack of expression of hormone receptors (e.g., estrogen receptor) makes TNBC more challenging to cure. As a result, there is still a deficiency of efficacious therapeutic interventions for TNBC [4]. Drug delivery approaches utilizing nanoparticles (NPs), i.e., nanomedicine, have shown that they have the potential to improve the drug accumulation in tumors and enhance antitumor efficacy. However, “off-target” expression and a low receptor density on cancer cells compromise the efficacy of ligand-based cancer targeting [5]. Various biomimetic strategies have recently been explored to target tumor cells and precisely deliver payloads to the desired tissues [6,7]. Coating NPs with the cell membranes from cancer cells endows them with an enhanced internalization and efficient tumor homing to syngeneic tumors [6,7]. Furthermore, cancer cell membrane (CCm)-coated NPs, such as mesoporous silica [8], metal–organic frameworks [9,10], and iron oxide [11], can evade immune clearance. 

Doxorubicin (DOX) is a commonly used chemotherapy agent for treating various types of cancer, including TNBC [12,13,14]. However, the use of DOX is associated with dose-dependent cardiotoxicity, which limits its clinical application [15]. To overcome this limitation, DOX can be encapsulated into NPs to improve its therapeutic index and reduce systemic toxicity [16]. Several studies have demonstrated the potential of DOX-loaded NPs, e.g., chitosan [17,18,19], poly(lactic-co-glycolic acid) [20,21], and mesoporous silica NPs [22,23], in the chemotherapy of TNBC. The unique properties of PSi NPs, such as their biodegradability, bioabsorbability, and high surface area for efficient drug loading, make them ideal for DOX loading [16,24,25]. Additionally, multimodal nuclear medicine imaging techniques, such as SPECT/CT (single-photon emission computed tomography/computed tomography) and PET/CT (positron emission tomography/computed tomography), have enabled the real-time monitoring of drug delivery in vivo and in vitro because of their remarkable precision and accuracy [26,27]. 

Based on the promising features of PSi for drug delivery, we hypothesize that CCm-coated PSi NPs are potential nanoplatforms that can be employed as biomimetic theranostics in imaging and targeting chemotherapy for TNBC. The present study employed a facile PEG-assisted CCm coating approach to develop radiolabeled and DOX-loaded biomimetic PSi NPs (Figure 1). Bisphosphonate (BP) molecules were conjugated on PSi as chelators for ^68^Ga and ^99m^Tc radionuclides. More importantly, the BP amount and pH effects were investigated to optimize the radiolabeling. The obtained biomimetic NPs possessed a favorable colloidal and radiochemical stability, a pH-dependent sustained drug release, homotypic targeting, and significant cytotoxicity against cancer cells in vitro. Given these promising features, these NPs hold great potential as theranostic platforms for tumor-targeting PET/CT or SPECT/CT imaging, along with targeting chemotherapy applications.

## 2. Materials and Methods

### 2.1. Chemicals and Materials

Silicon wafers were obtained from Okmetic (Vantaa, Finland). Hydrofluoric acid (HF 38–40%) and hydrochloric acid (HCl, 37%) were obtained from Merck (Rahway, NJ, USA). Hydrogen peroxide (H_2_O_2_) and ethanol (EtOH 99.5%) were purchased from ACROS Organics^TM^ (Geel, Belgium) and Altia Oyj (Helsinki, Finland), respectively. Fetal bovine serum (FBS), mesitylene (99%), 4,6-diamidino-2-phenylindole (DAPI), and EDTA-free mini protease inhibitor tablet were purchased from Sigma-Aldrich (St. Louis, MI, USA). Methoxy-PEG-silane (PEG, 0.5 kDa), methoxy-PEG-silane (PEG, 2 kDa), and DOX were purchased from Gelest (Morrisville, PA, USA), Biochempeg Scientific (Watertown, MA, USA), and Euroasias (Mumbai, India), respectively. The Bicinchoninic acid (BCA) assay kit and CellTiter-Glo assay kit were obtained from Thermo Fisher Scientific (Waltham, MA, USA) and Promega (Tokyo, Japan), respectively. Cyanine 5.5 NHS ester (Cy5.5) and 4-(2-hydroxyethyl)-1-piperazine ethane sulfonic acid (HEPES) were obtained from Lumiprobe (Cockeysville, MD, USA) and Biowest (Nuaillé, France), respectively. Dulbecco’s modified eagle medium (DMEM), phosphate-buffered saline (PBS, pH = 7.00 ± 0.02), Roswell Park Memorial Institute medium (RPMI, VWR), and Hank’s Balanced Salt Solution (HBSS) were purchased from VWR (West Chester, PA, USA). ^99m^Tc and ^68^Ga were freshly eluted from the radionuclide generator at Kuopio University Hospital. The synthesis of bisphosphonate has been described elsewhere [28]. All the chemicals and solvents were used without further purification.

### 2.2. Isolation of Cancer Cell Membranes

MAD-MB-231 (TNBC cell line) cells were cultured with RPMI medium supplemented with 10% FBS and 1% penicillin. Upon reaching 80% confluency, the cells were detached and washed with HBSS via centrifugation (1200× *g*, 5 min), then subjected to hypotonic lysing buffer and incubated at 4 °C for 30 min. The solution was then disrupted using a Dounce homogenizer in an ice bath with centrifugation at 3200× *g* for 5 min. After collecting the supernatant, hypotonic lysing buffer was added to redisperse the pellet. The homogenization and centrifugation steps were repeated. The resulting supernatants were collected and centrifugated at 6000× *g* for 20 min at 4 °C. The pellet was discarded and the supernatant was centrifuged at 100,000× *g* for 1 h at 4 °C to collect the pellets of the CCms. After being redispersed in 25 mM of HEPES, the concentration of cell membrane proteins was determined using a BCA assay kit, and the samples were stored at −20 °C for subsequent experiments. 

### 2.3. Preparation of Nanoparticles

(1) PSi NPs: The PSi films were fabricated using an electrochemical etching process with p-type silicon wafers (0.01–0.02 Ω cm) in a mixture of HF and ethanol in a 1:1 ratio. After etching, the PSi films were collected and dried at 65 °C for 1 h. Subsequently, the films were milled in ethanol using a planetary ball mill at 1000 rpm for 1 h. The NPs with the desired diameter around 160 nm were obtained through centrifugation (3500 rpm for 5 min) and subsequently stored in ethanol at ambient temperature for future use.

(2) BP-PSi NPs: The NPs were modified through the surface functionalization of the PSi NPs with various amounts of BP molecules. To eliminate the oxidized layer, a dispersion containing 10.0 g/L of PSi NPs (60 mL) was treated with 5% HF (6 mL) for 10 min. Following this, the PSi NPs were rinsed twice with ethanol and mesitylene. The PSi NPs then underwent a hydrosilylation process with various amounts (75, 150, and 300 mg) of BP molecules, employing a quartz tube under a N_2_ atmosphere at a temperature of 120 °C for 19 h. The resulting BP-PSi NPs were washed with ethanol and subsequently dispersed in ethanol for further characterization and analysis.

(3) ^99m^Tc-BP-PSi NPs: BP-PSi NPs, dispersed in 400 µL of ethanol at a concentration of 2.5 mg/mL, were introduced into a 1.5 mL glass ampoule equipped with a septum, followed by the replacement of the ambient atmosphere with nitrogen gas. A 300 MBq ^99m^TcO_4_^−^ solution was reduced with 100 µL of 1.0 mg/mL ascorbic acid and 100 µL of 1.0 mg/mL SnCl_2_. This mixture was then injected into each ampoule, resulting in a radioactivity level of approximately 7–8 MBq. The pH of the solution was adjusted using either 0.5 M of HCl or NaOH to reach different pH values (1, 3, 5, and 7). The ampoules were incubated at room temperature for 30 min. Later, the NPs were centrifugated (13,000 rpm for 10 min), washed twice with water, and ultimately dispersed in water for subsequent analysis. 

^68^Ga-BP-PSi NPs: the NPs were prepared by incubating 0.25 mg of BP-PSi NPs with approximately 3.5 MBq of ^68^GaCl_2_. The pH of the mixture was carefully adjusted using 0.5 M of NaOH to different pH values (1, 3, and 5), followed by an incubation period of 15 min at room temperature utilizing an End Over End Mixer. Next, the NPs were centrifugated, washed twice with water, and subsequently dispersed in water. 

The ^99m^Tc-BP-PSi and ^68^Ga-BP-PSi NPs exhibiting the highest radiochemical yields were selected for subsequent surface modification experiments.

(4) Cy5.5 labeled NPs: 2 mg ^99m^Tc-BP-PSi or ^68^Ga-BP-PSi NPs were initially subjected to the reaction with 10 µL of APTES. The amine-modified NPs were washed with ethanol and subsequently reacted overnight with a concentration of 0.25 mg/mL of Cy5.5-NHS at room temperature. A concentration of 5 mg/mL of succinic anhydride was used to cap the excess amine groups. The resulting NPs were obtained after additional washing with ethanol and served as the starting material for PEGylation and the CCm coating processes.

(5) PEG-^99m^Tc-BP-PSi or PEG-^68^Ga-BP-PSi NPs: the PEGylated NPs were prepared based on a previous report [24]. In the present study, PEG-silanes consisting of 0.5 kDa (100 µL) and 2 kDa (50 mg) were mixed with 1 mL of ethanol. This mixture was then combined with 1 mg of either ^99m^Tc-BP-PSi or ^68^Ga-BP-PSi NPs. Subsequently, the resulting mixture was incubated at 90 °C for 1 h under a controlled airflow. The PEGylated NPs were collected after washing with ethanol.

(6) Drug loading: 0.5 mg of PEG-^99m^Tc-BP-PSi or PEG-^68^Ga-BP-PSi NPs were mixed with 0.5 mg of DOX in PBS solution and then stirred at room temperature overnight. The mixture was centrifuged and washed twice with ethanol to remove the free drug. The supernatant was collected and measured with an ultraviolet-visible (UV-vis) spectrometer to obtain the DOX concentration based on the standard curve for DOX absorbance at 490 nm (Appendix A). Later, the DOX mass was calculated based on the volume of the supernatant. The obtained NPs were named as DOX-PEG-^99m^Tc-BP-PSi and DOX-PEG-^68^Ga-BP-PSi. The loading efficiency and loading degree were calculated according to the following equations:(1)$$Loading\;efficiency\;\left(\%\right)=\left(\frac{M_{i}-M_{SN}}{M_{i}}\right)\times 100$$
where *M_i_* is the initial DOX mass of the loading solution and *M_SN_* is the DOX mass in the collected supernatant after loading.
(2)$$Loading\;degree\;\left(\%\right)=\left(\frac{M_{i}-M_{SN}}{M_{NPs}}\right)\times 100$$
where *M_i_* is the initial DOX mass of the loading solution, *M_SN_* is the DOX mass in the collected supernatant after loading, and *M_NPs_* is the initial NPs mass used for the DOX loading.

(7) CCm-DOX-PEG-^99m^Tc-BP-PSi or CCm-DOX-PEG-^68^Ga-BP-PSi NPs: NPs were prepared using the solvent evaporation method. A total of 0.5 mg of DOX-PEG-^99m^Tc-BP-PSi or DOX-PEG-^68^Ga-BP-PSi NPs and 0.5 mg of CCm in 25 mM of HEPES were mixed and then dried under airflow at room temperature to evaporate the solvent. After removing the extra free CCm with centrifugation, CCm-coated NPs were obtained.

The radiochemical purity was measured using thin-layer chromatography (TLC). The acetone was regarded as the liquid phase. In total, 5 µL of NPs or free radioisotopes was spotted separately on TLC strips. Later, they were developed in acetone until the acetone front reached a specific distance from the bottom. The resulting position of the acetone front was used to calculate the radiochemical purity.
(3)$$Radiochemical\;purity\;\left(\%\right)=\left(\frac{R_{b}}{R_{b}+R_{t}}\right)\times 100$$
where *R_b_* is the radioactivity of the bottom of the strips measured with the gamma counter and *R_t_* is the radioactivity of the top of the strips measured with a gamma counter [29].

The radiochemical yield was calculated using the following formula:(4)$$Radiochemical\;yield\;\left(\%\right)=\left(\frac{R_{NPs}}{R_{isotope}}\right)\times 100$$
where *R_NPs_* is the radioactivity of the NPs measured after time correction based on the half-life of isotopes and *R_isotopes_* is the original radioactivity of the isotopes. All the radioactivity was determined using the gamma counter.

The radiochemical stability was evaluated in PBS and cell medium at 37 °C. The mixtures were sampled at different time intervals and centrifuged to collect the supernatant and NPs. Finally, the supernatant and NPs’ radioactivity were measured using the gamma counter, and the radiochemical stability was calculated as follows:(5)$$Radiochemical\;stability\;\left(\%\right)=\left(\frac{R_{NP}}{R_{SN}+R_{NP}}\right)\times 100$$
where *R_SN_* is the radioactivity of the supernatant and *R_NP_* is the radioactivity of the NPs. All the radioactivity was determined with the gamma counter.

### 2.4. Physicochemical Characterization 

The characterization of the NPs involved several techniques. The NPs’ morphology was investigated via transmission electron microscopy (TEM), employing a JEOL JEM2100F instrument. The zeta potential of the NPs was measured in deionized water at room temperature, while the hydrodynamic size of the NPs was determined in buffer solutions at 37 °C using a Zetasizer Nano ZS instrument (Malvern Instruments, Malvern, UK). The BP content of the NPs was evaluated with a thermogravimetric analysis (TGA (Canberra, Australia), NETZSCH (Selb, Germany)). The analysis was conducted under a nitrogen flow rate of 200 mL/min, involving equilibration at 80 °C for 30 min, followed by heating to 800 °C at a rate of 20 °C/min.

To verify the CCm coating, sodium dodecyl sulfate-polyacrylamide gel electrophoresis (SDS-PAGE) was employed. The DOX loading efficiency and release percentage were measured via ultra-visible absorption (490 nm) (UV–Vis, PerkinElmer Victor 3). The colloidal stability of the NPs over 24 h and their radiochemical stability were assessed in both PBS and RPMI cell medium (containing 10% fetal bovine serum and 1% penicillin) at 37 °C.

### 2.5. Drug Release

The release kinetics of the DOX from the PEGylated or CCm-coated NPs were investigated in Eppendorf tubes using pH 5.4 and pH 7 PBS buffers. To initiate the release process, the NPs were dispersed in a medium by applying ultrasound for approximately 5 s, followed by continuous rotation (24 rpm) at 37 °C. At predetermined time intervals, the NPs were separated via centrifugation, enabling the collection of the supernatant to analyze the released DOX. Subsequently, a fresh buffer solution was added to continue the release experiments. The collected samples were analyzed using UV-vis spectroscopy to assess the concentration of the released DOX and then calculate the released DOX mass with the total volume. All the samples had three replicates. The released DOX percentage was calculated according to the following equation:(6)$$Release\;DOX\;percentage\;\left(\%\right)=\left(\frac{M_{t1}+{M_{t2}+\cdots M}_{tn}}{M_{i}}\right)\times 100$$
where *M_i_* is the initial DOX mass of the DOX-loaded NP suspension and *M_t_* is the DOX mass of the supernatant collected at each time interval.

### 2.6. Cell Cytotoxicity

The cytotoxic properties of the free DOX, DOX-PEG-BP-PSi NPs, and CCm-DOX-PEG-BP-PSi NPs were evaluated using the CellTiter-Glo kit. The administered DOX doses were equal across all the experimental groups. MDA-MD-231 cells were seeded in 96-well plates at a density of 1 × 10^4^ cells per well and cultured in 100 μL of RPMI supplemented with 10% FBS and 1% penicillin-streptomycin. Subsequently, the culture medium was replaced with 100 μL of fresh medium containing various concentrations of free DOX, DOX-PEG-BP-Psi NPs, or CCm-DOX-PEG-BP-Psi NPs. The negative control consisted of cells cultured in the medium alone. Following incubation at 37 °C for 24 h, 48 h, and 72 h, cell viability was quantified using the CellTiter-Glo assay performed on a Synergy H1 microplate reader (Biotek, Winooski, VY, USA). All the samples had five replicates. 

### 2.7. Homotypic Targeting Studies

To investigate the hypothesis regarding the recognition ability of CCm-coated NPs by homotypic cancer cells, we performed a series of cellular experiments. First, MDA-MB-231, Raw 267.4, MCF-7, and HeLa cell lines were seeded in 8-well plates at a density of 2.5 × 10^4^ cells per well and cultured in 200 µL of cell medium for 24 h. Following this, the cell medium was replaced with a fresh medium containing NPs at a concentration of 50 μg/mL. The experimental groups consisted of CCm-DOX-PEG-^99m^Tc-BP-PSi or CCm-DOX-PEG-^68^Ga-BP-PSi NPs, while the control groups included free DOX, DOX-PEG-^99m^Tc-BP-PSi NPs, and DOX-PEG-^68^Ga-BP-PSi NPs. After a 4 h incubation period, the cells were washed three times with HBSS and fixed with a 4% paraformaldehyde solution for 10 min. Subsequently, the cells were washed twice with HBSS and incubated with DAPI solution (5 μg/mL in 200 µL of HBSS) for 10 min at room temperature to label the cell nuclei. The internalization of the NPs was visualized using a confocal laser scanning microscope (CLSM, Zeiss LSM 700, Carl Zeiss, Jena, Germany), where Cy5.5 was used to mark the PSi NPs (shown in red) and DAPI was used to stain the cell nuclei (shown in blue).

To further confirm the cellular internalization of the different NPs, TEM imaging was employed. Specifically, 2 × 10^5^ cells per well from various cell lines were seeded in a 24-well plate and cultured for 24 h. After a single wash with HBSS, the cells were treated with a medium containing NPs at a concentration of 50 µg/mL for 4 h. Subsequently, the cells were fixed with a solution of 2% glutaraldehyde and 0.1 M of PBS at 37 °C for 1 h. The cells were then washed twice with 0.1 M of PBS for 5 min, postfixed with 1% osmium tetroxide in 0.1 M of PBS for 1 h, and washed again twice with 0.1 M PBS for 5 min. The samples were prepared by cutting them into 60–80 nm frontal sections, followed by staining with uranyl acetate. TEM imaging was performed to observe and validate the homotypic targeting ability of the CCm-coated NPs.

## 3. Result and Discussion

### 3.1. The Effects of BP Amount and pH Value on Radiolabeling 

BP molecules act as chelators to bind both ^99m^Tc and ^68^Ga for SPECT/CT and PET/CT. To explore the effect of the BP amount on the radiolabeling, various mass ratios of BP to PSi NPs (0.125, 0.25, and 0.5) were used to prepare the BP-PSi NPs through a hydrosilylation process at 120 °C under a N_2_ flow for 19 h. The TEM images (Appendix A) showed that all the NPs presented similar irregular shapes. The dynamic light scattering (DLS) measurements (Appendix A) indicated that the mean diameters were 162 ± 1 nm (BP-PSi 1), 163 ± 1 nm (BP-PSi 2), and 166.0 ± 0.2 nm (BP-PSi). TGA curves (Figure 1a) revealed that a higher mass ratio (between the original BP and PSi NPs) led to a higher BP loading on the surface of the NPs, where the BP amounts were 0.87 wt% (marked as BP-PSi-0.125), 1.48 wt% (marked as BP-PSi-0.25), and 2.27 wt% (marked as BP-PSi-0.5), respectively. Compared to BP-PSi-0.125 (−30.7 ± 1 mV) and BP-PSi-0.25 (−37 ± 2 mV), the higher BP amount in the BP-PSi-0.5 NPs led to a more negative zeta potential (−39.6 ± 0.9 mV) because of the enhanced ionization of more phosphate groups (Figure 1b). During the radiolabeling, it became clear that the BP-PSi-0.5 NPs always had the highest radiochemical yield in the different pH conditions (Figure 1c,d); the maximum radiochemical yield of the BP-PSi-0.5 NPs reached 97% ± 2% (^99m^Tc, pH = 5) and 94.6% ± 0.2% (^68^Ga, pH = 5). Generally, all the NPs relied on the pH-dependent chelating mechanism, which can be explained by the deprotonation degree of the BP molecules. More specifically, four -OH groups exist on phosphonate moieties (Appendix A): the first pKa is ~1, the second pKa is ~3, and the third pKa is ~5 [30]. Notably, the ^99m^Tc radiolabeling was more sensitive to pH changes than ^68^Ga, potentially due to the reduction of ^99m^TcO_4_^−^ during the ^99m^Tc radiolabeling. Based on these findings, the BP-PSi NPs-0.5 NPs were chosen for the subsequent PEGylation, DOX-loading, and CCm coating. BP-PSi in the follow-up discussion was defaulted to BP-PSi-0.5.

### 3.2. Physiochemical Characterizations of PSi-Based Platform

Based on our previous work, PSi NPs were prepared using an electrochemical etching method [31]. BP molecules were then conjugated on the surfaces of the PSi NPs via a hydrosilylation reaction with Si-H groups to obtain BP-PSi NPs, in which the BP molecules chelated ^99m^Tc or ^68^Ga ions with a high affinity [24]. The shapes of the NPs were imaged with TEM measurements. Before the CCm coating, all the NPs were irregular without shape changes after the BP grafting, radiolabeling, PEGylation, and DOX loading (Appendix A). In comparison, an external layer was shown on the CCm-DOX-PEG-^99m^Tc-BP-PSi and CCm-DOX-PEG-^68^Ga-BP-PSi NPs, resulting from a successful CCm coating (Figure 2a–c). DLS measurements indicated that a separate increase of ∼10 nm in hydrodynamic diameter was observed after the PEGyaltion and CCm coating, confirming that both processes were successful (Figure 2d).

Furthermore, the zeta potentials (Figure 3a) showed that the BP-PSi NPs (−42.6 ± 0.2 mV) were more negative than the PSi NPs (−32.9 ± 0.8 mV). Later, successful radiolabeling and PEGylation led to a lower negativity of the as-formed NPs: ^99m^Tc-BP-PSi (−34.6 ± 0.9 mV), ^68^Ga-BP-PSi (−30.4 ± 0.5 mV), PEG-^99m^Tc-BP-PSi (−11.3 ± 0.9 mV), and PEG-^68^Ga-BP-PSi (−14 ± 1 mV). After positively charged DOX loading, the zeta potentials of the DOX-PEG-^99m^Tc-BP-PSi (1.8 ± 0.5 mV) and DOX-PEG-^68^Ga-BP-PSi NPs were both positive. The DOX molecules were loaded into the mesopores of the PEGylated NPs with a final loading efficiency of 35% ± 2%. Since the same masses of DOX and NPs were used during the loading process, the loading degree was 35% ± 2%, equivalent to 0.35 mg of DOX per milligram of NP. Owing to the success of the negatively charged CCm (−28.9 ± 0.5 mV) coating, the obtained CCm-DOX-PEG-^99m^Tc-BP-PSi (−20.9 ± 0.3 mV) and CCm-DOX-PEG-^68^Ga-BP-PSi (−24.6 ± 0.5 mV) NPs were highly negative in terms of their zeta potentials. The NPs presented a pH-responsive sustained release of DOX (Figure 3b): that is, more DOX was released at pH = 5.4 (tumor microenvironment) than at pH = 7 (physiological condition). This is attributed to the protonation of the amine groups in DOX at a lower pH. Under the acidic pH conditions, a higher DOX release with higher levels than that under physiological conditions was beneficial for obtaining an increased DOX concentration at tumor sites. Moreover, the CCm coating could further delay the release of DOX, indicating that it played the role of “gate-keeper”, reducing the undesired premature release of DOX (Figure 3b). SDS-PAGE was applied to examine the successful CCm coating on the NPs (Figure 3c). Similar bands on the CCm-coated NPs to those on the pure CCm indicated that the membrane proteins from the source cell membrane were well retained after the coating. 

The formed NPs were incubated in PBS and cell medium solution at 37 °C for 24 h to evaluate their colloidal stability. The negligible size changes in the two physiological conditions indicated that the long-term stability of the NPs was achieved after the PEGylation and CCm coating (Figure 3d,e). The radiochemical purity and yield of all the radiolabeled NPs were above 90%. The radiochemical stability was examined by incubating the NPs in PBS and cell medium for 24 h (^99m^Tc) or 2 h (^68^Ga) (Figure 3e,f). In PBS, all types of NPs showed a decreased trend in radiochemical stability over time. The CCm-coated NPs exhibited the highest radiochemical stability at 85.4% ± 0.9% (^99m^Tc, 24 h) and 88.7% ± 0.7% (^68^Ga, 2 h). Meanwhile, the pure NPs had the lowest radiochemical stability of 68.6% ± 0.7% (^99m^Tc, 24 h) and 75% ± 1% (^68^Ga, 2 h), because the pure NPs degraded faster without the protection from the layers of PEG and CCm. However, all the NPs displayed a relatively higher radiochemical stability in the cell medium than in PBS. One plausible explanation is that the heightened hydrolysis rate of the NPs in PBS relative to the cell medium resulted in a worse radiochemical stability.

### 3.3. In Vitro Homotypic Targeting Experiments and Cytotoxicity

The NPs were labeled with Cy5.5 to enable the study of their homotypic targeting ability with fluorescence microscopy. The physicochemical characterizations of the dye-labeled NPs involving the changes in size distribution and zeta potentials indicated a successful dye conjugation on the NPs (Appendix A) [32]. Next, the NPs with different surface modifications were incubated with four cell lines (MDA-MB-231 (Figure 4), MCF-7 (Appendix A), RAW 267.4 (Appendix A), and HeLa (Appendix A)). The PSi core and cell nuclei were labeled with Cy5.5 (red) and DAPI (blue), respectively, to monitor the intracellular distribution of the NPs. All the groups treated with the PEGylated NPs (DOX-PEG-^99m^Tc-BP-PSi and DOX-PEG-^68^Ga-BP-PSi) showed a weak and similar intensity of orange and red fluorescence signals from the DOX and PSi. The main reason was that the PEGylated NPs with DOX loading were positively charged, and a small portion of them experienced phagocytosis via electrostatic interaction after 4 h of incubation (Figure 4 and Appendix A). Obviously, the group treated with the CCm-coated NPs (CCm-DOX-PEG-^99m^Tc-BP-PSi and CCm-DOX-PEG-^68^Ga-BP-PSi) displayed the most robust red fluorescence, because the CCm coating endowed the NPs with homotypic targeting to the resource cancer cells (Figure 4). With the same treatments, weaker fluorescence signals of PSi and DOX were observed in the cell lines of MCF-7, RAW macrophages, and HeLa (Appendix A) compared to those in the MDA-MB-231 cells. This further supports the idea that the NPs with the CCm coating had homotypic targeting. 

TEM images (Figure 5) were further used to confirm the homotypic targeting properties of the CCm-coated NPs. The results were consistent with those shown in the CLSM images (Figure 4). Specifically, PEGylated NPs (DOX-PEG-^99m^Tc-BP-PSi, and DOX-PEG-^68^Ga-BP-PSi) were rarely observed inside all cells. However, both CCm-DOX-PEG-^99m^Tc-BP-PSi and CCm-DOX-PEG-^68^Ga-BP-PSi had an enhanced intracellular uptake, and most of the NPs were located inside the endocytic vesicles of the MDA-MB-231 cell lines. On the contrary, a very minor amount of the NPs was found in other cell lines. 

The samples were incubated with the MDA-MB-231 cell line for different time periods (24 h, 48 h, and 72 h) to evaluate their targeting therapeutic effect (Figure 6). With an increase in the concentration of DOX and incubation time, all the groups showed a significantly increased cytotoxicity. The DOX-PEG-BP-PSi and CCm-DOX-PEG-BP-PSi NP-treated groups revealed an increased cytotoxicity compared to the free DOX group at 24 h, 48 h, and 72 h. This phenomenon could be attributed to the DOX-PEG-BP-PSi NPs being positively charged and interacting with the negatively charged cancer cell membranes via electrostatic adsorption. As a result of the homotypic targeting recognition of CCm, the CCm-DOX-PEG-BP-PSi NPs were the most effective at killing cancer cells among these groups. Additionally, the therapeutic effect was evaluated with the Raw267.4 and MCF-7 cell lines treated with the CCm-DOX-PEG-BP-PSi NPs (Appendix A). The biomimetic NPs displayed the highest cytotoxicity for MDA-MB-231 in 1.0 μg/mL of DOX, but the highest cytotoxicity for Raw 267.4 in 2.5 μg/mL of DOX. The possible reason is that the Raw 267.4 cell line is more sensitive than the cancer cell line at higher DOX doses [33]. These results revealed that the CCm-DOX-PEG-BP-PSi NPs could have positive therapeutic effects in targeting chemotherapy for MDA-MB-231 cancer. 

## 4. Conclusions

In this study, we investigated the influence of BP amount and pH value on the ^99m^Tc and ^68^Ga radiolabeling process of BP-grafted PSi NPs. Our findings revealed that increasing the BP amount in the NPs resulted in a more negative zeta potential and significantly enhanced the radiolabeling yield under various pH conditions. Furthermore, the enhanced deprotonation degree of the BP molecules at higher pH levels corresponded to an improved radiolabeling yield. As a result, we selected the optimized NPs with the maximum 2.27 wt% BP amount for subsequent modifications, including PEGylation, DOX loading, and CCm coating. Our analyses confirmed that surface modifications of the NPs were successful, giving a good colloidal stability in both PBS and cell medium. Notably, the CCm-coated NPs exhibited a superior radiochemical stability (^99m^Tc: ≥85.4% ± 0.9%, 24 h and ^68^Ga: ≥88.7% ± 0.7%, 2 h) compared to the other NPs, indicating the protective effect of the CCm layer. In vitro cytotoxicity experiments demonstrated that the CCm-coated NPs displayed an enhanced cytotoxicity and targeted therapeutic efficacy in the MDA-MB-231 cancer cells relative to other groups, verifying the targeting ability of biomimetic NPs from the cell membrane coating. Fluorescence microscopy and TEM images further validated the homotypic targeting and internalization capability of the CCm-coated NPs in the MDA-MB-231 cells. This finding highlights CCm-coated NPs’ promising applications in targeting triple-negative breast cancer imaging and therapy, which can lead to exciting opportunities for further preclinical investigation and utilization.

## Data Availability

Not applicable.

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
