# Peer review of "Biomimetic Inorganic Nanovectors as Tumor-Targeting Theranostic Platform against Triple-Negative Breast Cancer"

_pharmaceutics, 2023, doi:10.3390/pharmaceutics15102507_

Round 1

Reviewer 1 Report

 The paper is on biomimetic inorganic nano vector as tumor-targeting theranostic platform against triple2 negative breast cancer.

Although paper sounds scientific it has severe flaws that I dont ththink it would be accptable to be published in this journal.

the experiments have been only in vitro and not any in vivo study

the conclusion is too short

the paper format is not siutable for this journal

Author Response

In consistent with your point of view, we also believe that although the performance of the biomimetic nanovector in vitro is very good, it could be better if it could be extended to in vivo experiments. However, successful in vitro experiments are the prerequisite for in vivo experiments. The present study shows that chelator amount and pH conditions play an important role in the radiolabeling process, and the cancer coating method endows the nanovector with the targeting ability for enhanced chemotherapy. These findings conclude that this type of nanovector has great potential for targeting therapy of triple-negative breast cancer in vivo. However, in vivo experiments were not in the scope of the present study. We will carry out in vivo animal tests in the future.

We have added some text with yellow highlights on page 22 in the conclusion to make it more informative.

We thank the reviewer for the comment related to the format of the manuscript. The current version of the manuscript was submitted for scientific review only. The format of the manuscript will be updated according to the requirements of the journal after it is accepted.

Reviewer 2 Report

The cover of material was done with cancer cell membrane proteins and not with cancer cell membrane proteins, thus please correct in all text. Membranes are composed of lipids where proteins of membrane are embedded thus, I am wondered if lipids from membranes are also part of the composition of the material used to cover the NPs?

Figure 2 legend “…. d) Mean 273 diameter of all types of NPs and pure MDA-MB-231 CCm.” Thus, it means that the cancer cell membrane protein is aggregated proteins or micelles?

With this size would be expectable that in the TEM images be observed?

The stability was done in RPMI base or complete medium containing serum? Please specify it in material and methos.

To confirm the specificity of the particles when they have adsorbed cell proteins, it is necessary to present cytotoxicity studies with other cells, particularly non-cancerous cells such as MCF10A.

Author Response

Reviewer # 2:

Comment 1: The cover of material was done with cancer cell membrane proteins and not with cancer cell membrane proteins, thus please correct in all text. Membranes are composed of lipids where proteins of membrane are embedded thus, I am wondered if lipids from membranes are also part of the composition of the material used to cover the NPs?

Answer: We thank the reviewer for the critical comment. The coating on the NPs was cancer cell membranes composed of proteins and lipids, which was verified in our previous work (Nature Communications 2022, 13, 6181).

Comment 2: Figure 2 legend “…. d) Mean diameter of all types of NPs and pure MDA-MB-231 CCm.” Thus, it means that the cancer cell membrane protein is aggregated proteins or micelles?

With this size would be expectable that in the TEM images be observed?

Answer: We thank the reviewer for the critical comment. First, we need to clarify that the coating on the NPs was cancer cell membranes, which are composed of membrane proteins and lipids. Based on the dynamic light scattering measurement, the average size of pure cancer cell membranes is 142±3 nm. This size could be easily observed by transmission electron microscopy in Figure 2a.  

Comment 3: The stability was done in RPMI base or complete medium containing serum? Please specify it in material and methods.

Answer: Thanks for the comments. The stability was done in RPMI cell medium containing 10% fetal bovine serum and 1% penicillin with yellow highlights on lines 16-17, page 9.

Comment 4: To confirm the specificity of the particles when they have adsorbed cell proteins, it is necessary to present cytotoxicity studies with other cells, particularly non-cancerous cells such as MCF10A.

Answer: To address this comment, we have tested the cytotoxicity of CCm-DOX-PEG-BP-PSi NPs with MCF-7 (cancerous cells) and Raw 267.4 (non-cancerous cells) cell lines. The results in newly added Figure S9 indicate that biomimetic NPs (1.0 μg/mL DOX) have the highest cytotoxicity for MDA-MB-231. Meanwhile, biomimetic NPs (2.5 μg/mL DOX) have the highest cytotoxicity for Raw 267.4, the possible reason is that the Raw 267.4 cell line is more sensitive than the cancer cell line (Biomacromolecules 2011, 12, 3612). The texts have been added with yellow highlights in lines 5-9, page 21. 

Reviewer 3 Report

Wen at al have reported on development of an inorganic targeted nanomedicine for breast cancer treatment. Although, the use of mesoporous silicon nanoparticles as drug carrier has been extensively explored. The authors have proposed dual functionality that could be of an interest to a wide range of readers of this journal. Before suggesting this manuscript for publication, there are a few issues that need to be addressed.

1.       The abstract is very technical, it is suggested that the authors revise the abstract and focus rather on the outcome of this research.

2.       the introduction could be further strengthening by providing more information on the use of silicon NPs in nanomedicine such as doi.org/10.1021/acsami.6b14836.

3.       could the authors comment of cell uptake pathway of these CCm-DOX-PEG-68Ga-BP-Psi since the average particle size is quite large for cell internalization?

4.       In Fig.4 first row on the right-hand side, why do the Free Dox NPs fluoresce? Also, why Cy5.5 is not visible in the first 3 rows?

5.       Could the authors explain why do the Free Dox show even at low dosage show therapeutic effect with less than 20% cell variability?

Author Response

Reviewer # 3:

Wen at al have reported on development of an inorganic targeted nanomedicine for breast cancer treatment. Although, the use of mesoporous silicon nanoparticles as drug carrier has been extensively explored. The authors have proposed dual functionality that could be of an interest to a wide range of readers of this journal. Before suggesting this manuscript for publication, there are a few issues that need to be addressed.

Answer: We appreciate the positive feedback for our study.

Comment 1: The abstract is very technical, it is suggested that the authors revise the abstract and focus rather on the outcome of this research.

Answer: The abstract has been revised with yellow highlights on pages 1-2 based on the comment.

Comment 2: the introduction could be further strengthening by providing more information on the use of silicon NPs in nanomedicine such as doi.org/10.1021/acsami.6b14836.

Answer: The reference has been cited with yellow highlights on line 8, page 3, in the introduction part.

Comment 3: could the authors comment of cell uptake pathway of these CCm-DOX-PEG-68Ga-BP-PSi since the average particle size is quite large for cell internalization?

Answer: We thank the reviewer for the critical comment. A recent study has shown that both cancer cell membrane-coated PSi or naked PSi NPs mainly follow clathrin-mediated endocytosis and micropinocytosis (Adv. Healthcare Mater. 2020, 2000529), where the size of NPs is under the range from 180 to 200 nm close to the average particle size of CCm-DOX-PEG-68Ga-BP-PSi (197±4 nm).

Comment 4: In Fig.4 first row on the right-hand side, why do the Free Dox NPs fluoresce? Also, why Cy5.5 is not visible in the first 3 rows?

Answer: We thank the reviewer for the critical comment. DOX has a maximum emission wavelength of 560 nm (Reactive Oxygen Species (Apex). 2016, 2, 432–439), causing the fluorescence in Figure 4. Cy5.5 serves as the dye for labeling the PSi core. In the first row, the cells are treated with free DOX. In the second and third rows, the cells are treated with PEGylated PSi NPs. The PEGylation could result in decreased cellular internalization with negligible Cy5.5 fluorescent signals (Journal of Biomedical Nanotechnology 1, 2005, 397-401). The non-internalized NPs have been washed away with buffer solutions before taking confocal images. Thus, Cy 5.5 was not visible in the first 3 rows.

Comment 5: Could the authors explain why do the free Dox show even at low dosage shows therapeutic effect with less than 20% cell variability?

Answer: We thank the reviewer for the critical comment. DOX is a small drug with a positive charge due to amine groups in its molecular structure. Free DOX can easily pass through the cell membrane of cells to cause DNA damage (Johnson-Arbor, K., & Dubey, R. (2017). Doxorubicin). Thus, the cell viability was less than 20% after 72 h incubation even though a low dosage of DOX was used. However, the use of free DOX presents non-specific distribution to damage healthy tissues such as the heart (International journal of pharmaceutics 554, 327-336). So, the use of NPs as the carrier is needed to enhance therapy efficacy and decrease side effects.

Round 2

Reviewer 1 Report

I think the paper is ok to publish in Pharmaceutics at the current shape. 

In vivo studies are needed for completing the study though.

Author Response

We thank the reviewer for the positive but critical feedback. Our current results in vitro already demonstrated the potential of our biomimetic inorganic nanovector in targeting chemotherapy of triple-negative breast cancer. Our aim was to report the results supporting the future in vivo experiments in the present paper and, as suggested by the reviewer, to report the in vivo results in the next paper.

Reviewer 2 Report

As no non-cancerous cell line from the same tissue as the MDA-MB-231 cell line was tested and chosen non-cancerous RAW 264.7 Cell Line murine, a macrophage cell line tested, the authors have no experimental support based on the cytotoxicity results to support the following sentence in the summary ”The in vitro studies demonstrated that the biomimetic NPs exhibited enhanced cellular uptake and increased delivery of DOX to cancer cells because of their homotypic targeting ability, resulting in better chemotherapy than free DOX or pure NPs. Altogether, these findings indicate the potential of the developed platform in cancer treatment and diagnosis.”

The specificity and targeting are not completely supported by the results, so the authors must reformulate the text.

Author Response

We thank the reviewer for the critical comment. The targeting ability of NPs has been evaluated by incubating the NPs with cancerous cell lines: MDA-MB-231 (breast cancer), MCF-7 (breast cancer), HeLa (cervical cancer), and non-cancerous cell line: RAW 264.7. The results in CLSM and TEM have verified the homotypic targeting ability of biomimetic inorganic nanovector. However, as suggested by the reviewer, the text has been reformulated in the new version of the abstract (page 2, lines 2-4): The in vitro studies demonstrated that the biomimetic NPs exhibited enhanced cellular uptake and increased delivery of DOX to cancer cell, resulting in better chemotherapy than free DOX or pure NPs. 

Round 3

Reviewer 2 Report

The authors consider all observations made to the manuscript improving it. Thus, in my opinion, the article may be considered for publication.